# Study on Acoustic Emission and Coda Wave Characteristics of Layered Cemented Tailings Backfill under Uniaxial Compression

**Chongjie Huang** [1,2]**, Wen He** [1,2,3,*] ID**, Bokai Lu** [1,2]**, Manman Wang** [1,2]**, Shenhai Li** [1,2] **and Changbo Xiao** [4]

1   School of Resources and Environmental Engineering, Jiangxi University of Science and Technology, Ganzhou 341000, China; 6120200007@mail.jxust.edu.cn (C.H.); 6720210995@mail.jxust.edu.cn (B.L.); 6720211067@mail.jxust.edu.cn (M.W.); jxust_lsh1995@163.com (S.L.)
2   Jiangxi Province Key Laboratory of Mining Engineering, Jiangxi University of Science and Technology, Ganzhou 341000, China
3   Engineering Research Center for High-Efficiency Development and Application Technology of Tungsten Resources, Jiangxi University of Science and Technology, Ministry of Education, Ganzhou 341000, China
4   Jiangxi Zhongliang Blasting Engineering Co., Ltd., Pingxiang 337000, China; xiao13807997583@163.com
*   Correspondence: hewen@jxust.edu.cn

**Abstract:** The paper analyzes the effects of filling times and filling interval time on the acoustic emission characteristics and coda wave characteristics of layered cemented tailings backfill under uniaxial compression and, to a certain extent, enriches the study of layered cemented tailings backfill in this field. The work aims to monitor the early warning of layered cemented tailings backfill with different layering factors during deformation and damage by the changing law of acoustic emission and ultrasonic signals. By conducting uniaxial compression tests, acoustic emission, and ultrasonic tests of layered cemented tailings backfill, the acoustic emission parameters and their fractal characteristics of layered cemented tailings backfill with different layering factors during uniaxial compression were calculated. Meanwhile, the variation law of the coda wave velocity variation rate of layered cemented tailings backfill during uniaxial loading was analyzed using coda wave interferometry. The test results show the feasibility of using acoustic emission and ultrasonic means to monitor and warn about the deformation damage of layered cemented tailings backfill.

**Keywords:** layered cemented tailings backfill; acoustic emission; fractal dimension; coda wave interferometry; uniaxial compression

## 1. Introduction

Underground mining is one of the main ways to develop and utilize metal mining resources worldwide. However, the filling mining method is one of the main underground mining methods, and the filling mining method is currently the closest to the goal of green and efficient safe mining [1–3]. To improve economic benefits, the same goaf will be filled with different concentrations of filling slurry so that large-scale goaves on the current filling capacity cannot achieve a one-time completion, resulting in an apparent layered structure of the filling body. As the filling body has an essential impact on the stability of the stope and the safety of mine production [4], it is of great engineering significance to monitor and warn of the deformation and damage of the layered cemented tailings backfill (LCTB) using acoustic emission and ultrasonic signal characteristics.

Currently, the research field of LCTB mainly focuses on the backfill's mechanical properties and failure mode. Zhao et al., studied the mechanical properties and synergistic deformation characteristics of combined cemented tailings backfill with different cement tailings ratios as variables [5]. Cao et al., investigated the influence of structural factors on the variation law of compressive strength and failure mechanism of cemented tailings backfill and explored the mechanical properties of LCTB [6]. Wang et al., studied the

mechanical properties and acoustic emission characteristics of LCTB from different cement tailings ratios and height ratios [7,8]. However, using acoustic emission and ultrasonic means is more to study the filling body of a complete structure [9–11]. Cheng et al., studied the spatiotemporal evolution law of acoustic emission parameters during the compression process of the filling body and established a rupture prediction model by combining the cusp mutation theory [12]. Gong et al., studied the filling body's acoustic emission and b-value characteristics under the loading and unloading conditions and discovered the energy evolution distribution law of the filling body during the loading and unloading process [13]. Zhao et al., summarized the relationship between acoustic emission event rate, ringing count rate, and stress time of tantalum–niobium tailings backfill in uniaxial compression and splitting tests and analyzed the stress and acoustic emission characteristics of each stage [14]. Cheng et al., established a prediction model for the strength of cemented tailings backfill, ultrasonic wave velocity, and density of the backfill in the ultrasonic testing and mechanical tests of the cemented tailings backfill and constructed the damage evolution model of cemented tailings backfill [15]. He et al., used ultrasonic waves to monitor the strength of the filling body and established the relationship between the strength of the filling and the parameters of the guided wave [16]. Yan et al., characterized the early-age behavior of cemented tailings backfill by the ultrasonic intensity and spectrum [17].

However, there is a lack of research results on the use of acoustic emission and ultrasonic signal characteristics to monitor and warn about the deformation damage of LCTB. Therefore, this paper performs uniaxial compression acoustic emission tests and ultrasonic tests on LCTB based on previous studies on the mechanical properties of LCTB [18–21] and the acoustic emission and ultrasonic characteristics of intact cemented tailings backfill [22–25]. Two factors—different filling times and filling interval time—are considered to analyze the changes in acoustic emission and ultrasonic signals during the deformation and damage of LCTB. Furthermore, it demonstrates the feasibility of using acoustic emission and ultrasonic means to monitor and provide a theoretical basis for the early warning of deformation damage of LCTB.

## 2. Materials and Methods

### 2.1. Materials

The graded tailings used in the tests were from a copper mine in Jiangxi Province, China. The tailings were dried and analyzed for particle size and the particle size distribution is shown in Figure 1. According to the data analysis, the particle size range is mainly distributed from 0.8 μm to 300.0 μm, with an average particle size of 150.4 μm and a median particle size of 107.1 μm. Through the calculation, the inhomogeneity coefficient is 2.65, the curvature coefficient is 1.13, the tailings' particle gradation is widely distributed, and the particle grade distribution is concentrated. In this test, the XRD-2700 diffractometer was used for X-ray diffraction testing of the tailings; the X-ray diffraction pattern of the tailings is shown in Figure 2. The analysis by JADE 6.0 software (Materials Data, Inc., Livermore, CA, USA) shows that the main composition of the tailings is $SiO_2$. Conch brand ordinary Portland cement (P.O.42.5) was selected as the cementitious material, and the backup water for the filling system was ordinary tap water.

### 2.2. Methods

For the mine filling process and the actual filling engineering, the slurry concentration of the filling body prepared in the test was 70%. The following two factors determine the delamination of the cemented tailings backfill for the test:

(1) Delamination caused by the number of filling times in the same goaf. The number of filling times in this test was set to 1, 2, 3 times, the interval time of each filling was 12 h, and the cement tailings ratio of the prepared LCTB was 1:8. For specimens filled twice, the height of each filling is one-half of the height of the mold, and for specimens filled three times, the height of each filling is one-third of the height of the mold.

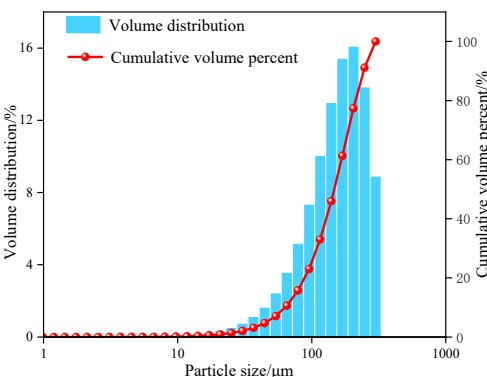

**Figure 1.** Particle size distribution of the tailings.

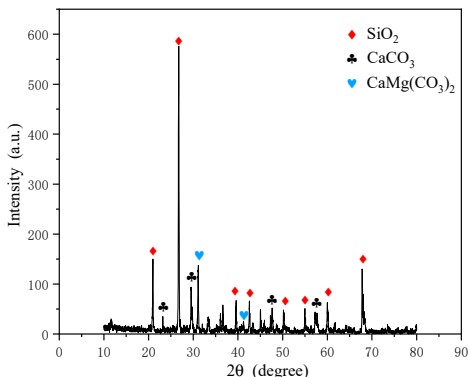

**Figure 2.** X-ray diffraction.

(2) Delamination caused by filling interval time. The filling interval time considered in the tests were mainly 0 h, 12 h, and 24 h. The prepared specimens were all LCTB with two filling times, the cement-to-tailings ratio of the first-stage filling slurry was 1:4 and the cement tailings ratio of the second-stage filling slurry was 1:8.

Firstly, mix cement, tailings, and water according to the above ratio, then mix them in a JJ-5 mixer for 5 min until all materials are mixed evenly, and finally put them into the mold. The test was conducted using a cylindrical transparent acrylic tube with an inner diameter of 50 mm and a height of 100 mm as the mold of specimens. After specimens of LCTB were made and molded for 24 h, they were demolded. Then, all the specimens were put into a maintenance box. The relative humidity was not less than 95%. The relative temperature was set at $(20 \pm 5)$ °C and the maintenance period was 28 days. The test process is shown in Figure 3.

The uniaxial compression acoustic emission test was carried out on the RMT-150C rock mechanics test system with the control method of stroke control and the set displacement loading rate of 0.02 mm/s [26]. The Micro-II Digital AE System is used for acoustic emission signal acquisition during the loading process. Two acoustic emission sensors were arranged in the middle of the vertical height of each specimen. The type of sensor used for this acoustic emission signal acquisition was UT1000, which has a resonant frequency of 60~1000 kHz, a preamplification of 40 dB, a threshold voltage value of 35 dB, and a sampling rate of 1 MSPS.

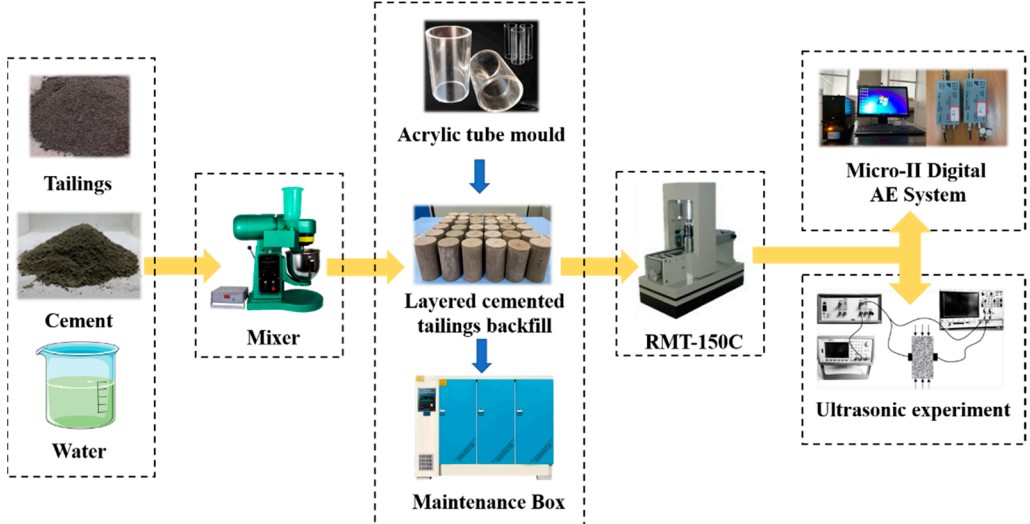

**Figure 3.** Schematic diagram of the experimental process.

The mechanical equipment used for the uniaxial multi-stage loading ultrasonic test was still the RMT-150C rock mechanics' test system with force-displacement control. In order to ensure the validity of the data, each target stress value was set according to the maximum compressive strength of the different types of specimens, and the maximum loading force of this test did not exceed 75% of the peak stress of the specimen; specific target stress values are shown in Table 1. At the same time, ultrasonic testing was performed on the specimens with the help of an ultrasonic testing system to record the ultrasonic signal changes of the LCTB under different stresses. The ultrasonic testing system consists of a waveform generator, power amplifier, oscilloscope, and sensor; the specific equipment models are shown in Table 2.

**Table 1.** Target stress value of the LCTB.

| Factors | Variables | Target Stress Value/N |
|:---:|:---:|:---:|
| filling times | 1 | 1570 |
| | 2 | 1197 |
| | 3 | 1000 |
| filling interval time | 0 h | 1943 |
| | 12 h | 1629 |
| | 24 h | 1256 |

**Table 2.** Equipment information for ultrasound systems.

| Equipment Name | Waveform Generator | Power Amplifier | Oscilloscope | Sensor |
|:---:|:---:|:---:|:---:|:---:|
| **Model** | 33522B | 2350 | DSOX2022A | R6$\alpha$ |

## 3. Results

### 3.1. AE Characteristics

3.1.1. Analysis of Acoustic Emission Parameters

Figures 4 and 5 show the relationship curves of the ringing count, energy, and time of the LCTB with different filling times, respectively. Figure 4 shows that the changes in the ringing counts of LCTB specimens with different filling times during loading are the same as the overall trend of stress changes with time. With the increasing filling times, the peak stress of the LCTB specimens decreases, the peak ringing count decreases, and the

cumulative ringing count also decreases. The main reason is that as the filling times of the LCTB increase, they increase the overall internal pore space of LCTB. The internal pore space of the LCTB with fewer filling times is compressed earlier during the loading process, thus generating new cracks; hence, its ringing count is more active and the cumulative ringing count is significant. From Figure 5, it can be seen that the peak energy of the LCTB increases with the filling times. In the early loading stage, the ringing count and energy of the three types of LCTB specimens all show a certain degree of sudden increase, and the peak value of ringing count and energy appeared earlier than the stress peak. The difference is that in the later stage of the stress peak, the ringing count is in a quiet state, while the energy shows several sudden increases. Thus, the sudden increase in ringing count and energy can be regarded as the precursor of instability of the LCTB.

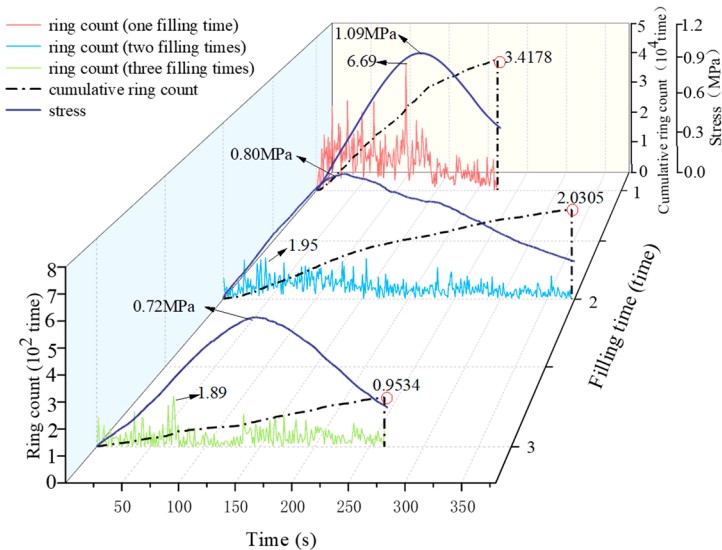

**Figure 4.** Ringing count of LCTB with different filling times.

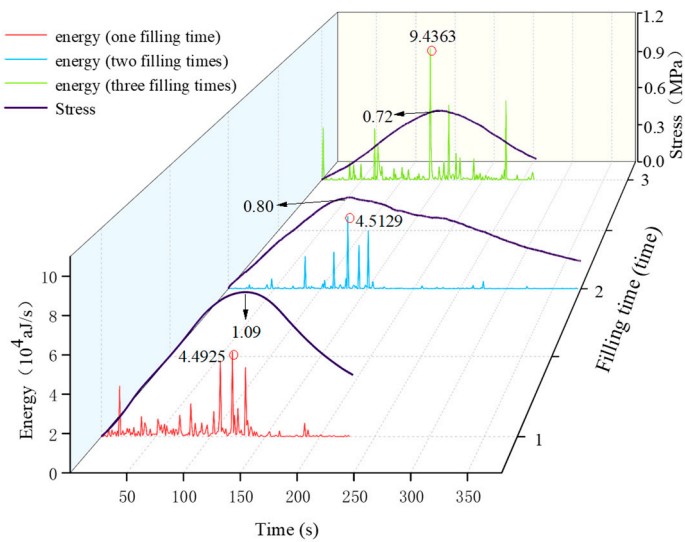

**Figure 5.** The energy of LCTB with different filling times.

Figures 6 and 7 show the relationship curves of the ringing count, energy, and time for LCTB with different filling interval times, respectively. The relationship between ringing count, energy, and time for three different filling interval time specimens is very similar. As the filling interval time increases the peak stress of the LCTB decreases and the peak ringing count and peak energy decrease. In the initial loading stage, many pores inside the LCTB are compacted, all three types of LCTB show an active period of acoustic emission,

and the ringing count and energy both increase sharply during this period. In the linear elastic stage, the internal pores of the LCTB are further compacted and a small number of tiny cracks sprout, the acoustic emission activity is more intense, and all three types of LCTB show a surge point. Comprehensive analysis shows that with the extension of the filling interval time, the gap between the LCTB sublevels increases, resulting in more internal particle friction during uniaxial compression and more acoustic emission signal burst points.

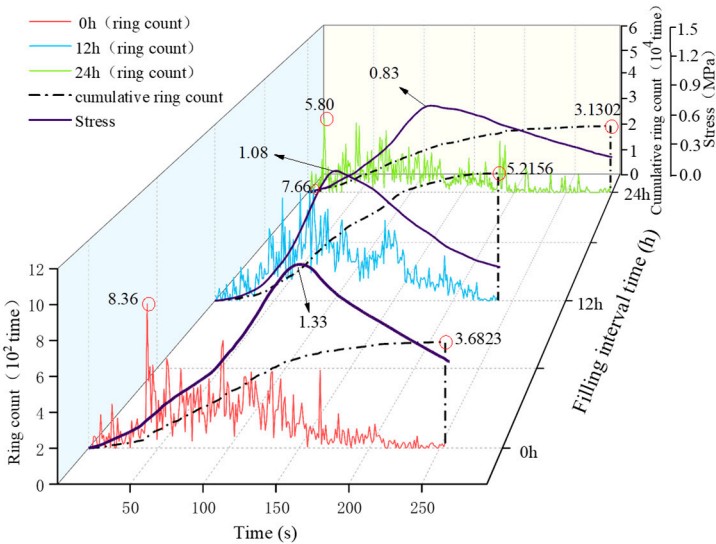

**Figure 6.** Ringing count of LCTB with different filling interval time.

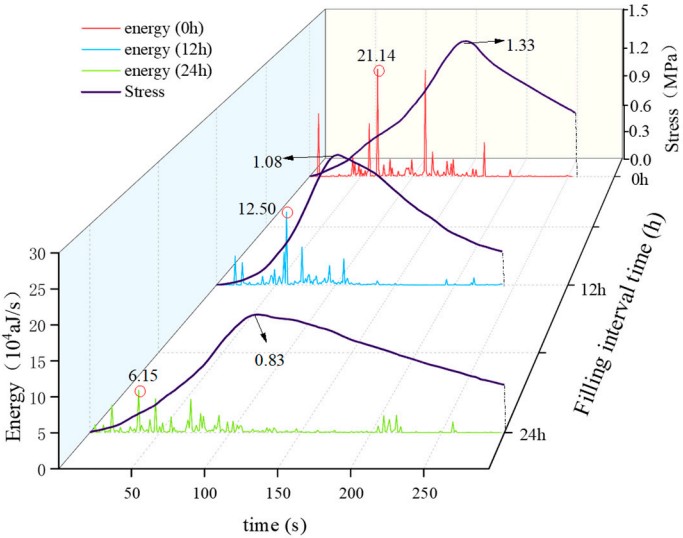

**Figure 7.** The energy of LCTB with different filling interval time.

### 3.1.2. Fractal Characteristics of Ringing Counts

The correlation dimension applies to calculating the dimension of acoustic emission parameters and time series. In the theory of the correlation dimension G-P algorithm, the following two formulas are the main ones [27,28]:

$$W[r(k)] = r(k)^D \tag{1}$$

$$D(m) = \ln W[r(k)] / \ln r(k) \tag{2}$$

where $W[r(k)]$ is the correlation function; $r(k)$ is to the measurement scale and $k$ is the scale factor; $D$ is the correlation dimension; $m$ is the phase space dimension.

In order to determine a reasonable phase space dimension m for the ringing counting parameter, a series of phase space dimensions are selected for analysis, as shown in Figure 8. The figure shows that the change of the $D$ value tends to be stable when $m = 3$, so m is taken as 3 for the calculation of the association dimension.

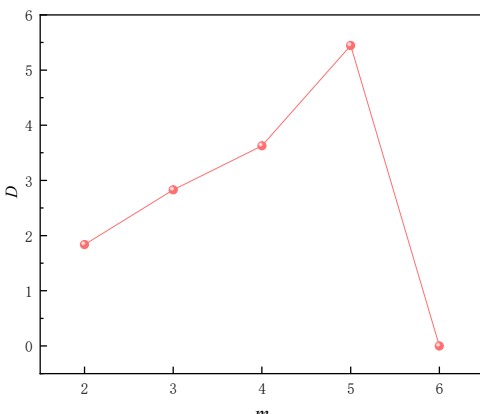

**Figure 8.** The relationship between phase space dimension m and correlation dimension $D$.

In order to study the fractal characteristics of the ringing counts of LCTB with different layering factors, the correlation dimension $D$ values of various types of LCTB specimens at different stress levels were calculated. Figure 9 shows the fractal curve of the ringing count of the LCTB with different filling times. The figure shows that the minimum value of the correlation dimension $D$ appears later as the filling times increase. The change process of the correlation dimension $D$ value of the ringing count for all three types of LCTB can be roughly divided into three stages: up–down–up. Before 30% stress, the correlation dimension $D$ values of the three types of LCTB are all at relatively high levels, indicating that the specimens are in a relatively disordered state at this time; this is because the LCTB specimens are in the compacting stage and the primary internal fractures are compacted. As the stress increases to about 50% of the peak stress, the pores are densified and new cracks begin to sprout; the orderliness of the LCTB increases and the $D$ value decreases. Among them, the $D$ value reaches the minimum value at the peak stress level of 50%–60% for the LCTB specimen with one filling time and the minimum $D$ value at the peak stress level of 60%–70% for the LCTB specimen with two filling times, and the minimum $D$ value at the peak stress level of 70%–80% for the LCTB specimen with three filling times. During the experiment, the orderliness of the specimens further increases and larger cracks are about to appear and produce macroscopic rupture.

Figure 10 shows the fractal curves of the ringing count of the LCTB at different filling interval times. The figure shows that the minimum value of the correlation dimension $D$ value of the LCTB ringing count is smaller as the filling interval time increases. The change process of the correlation dimension $D$ value of the ringing count for the LCTB with three different filling interval time can be roughly divided into three stages: down–up–down. Before the peak stress level of 25%, due to the inconsistent strength of the upper and lower parts of the LCTB, the primary cracks are compacted tightly and a small number of new cracks are initiated. This stage is mainly dominated by small-scale damage within the LCTB and the $D$ value decreases. With the further increase of the load, the cracks inside the LCTB diffracted and interleaved at around 70% of the peak stress level and the $D$ values all appeared to be relatively large, at which time the specimens are in disorder. Approaching the peak stress, the new cracks inside the filling body intertwine with the existing cracks and produce the crack extension phenomenon; the $D$ value decreases and the orderliness increases.

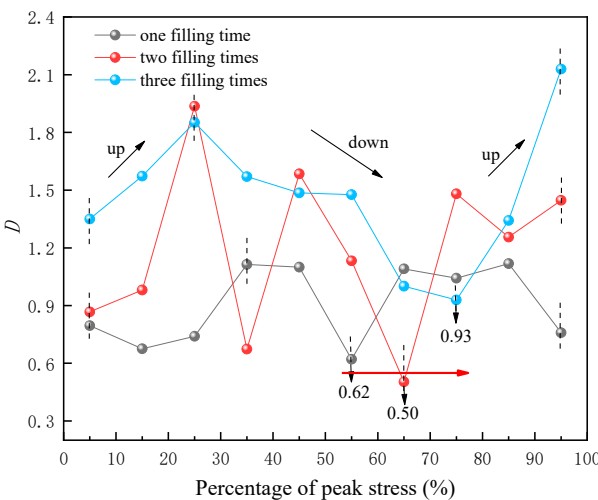

**Figure 9.** *D* of LCTB with different filling times.

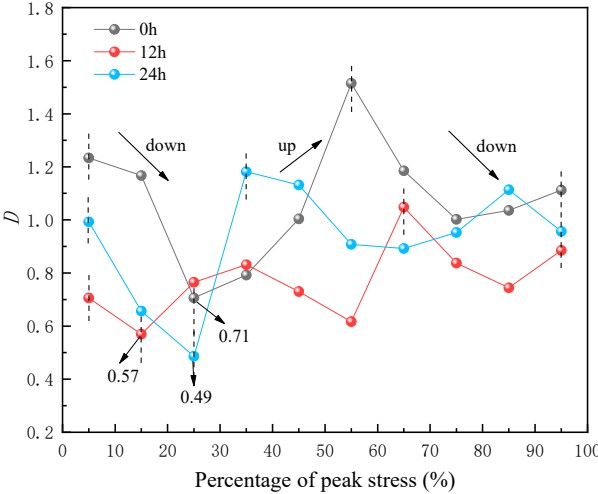

**Figure 10.** *D* of LCTB with different filling interval time.

### 3.2. Analysis of Coda Wave

#### 3.2.1. Coda Wave Interferometry and Ultrasonic Parameter Determination

Coda wave interferometry uses the coda wave formed by multiple scattering to study the medium, mainly to study the rate of change of the coda wave velocity [29,30]. Since the coda wave interferometry method was proposed, there are mainly two data processing methods: the moving split-windows method and the trace stretching method. Combined with the research of many scholars, the trace stretching method is finally selected as the analysis method of this paper. The idea of the trace stretching method is to take a waveform at a certain state of the medium as a reference waveform and then cross correlate the waveforms at each state with the reference waveform to calculate the rate of change of the coda wave velocity. It corresponds to the maximum correlation coefficient by stretching or compressing the time axis of the waveform, as shown in Figure 11 [31,32].

The correlation coefficient is defined as [32]:

$$F(\varepsilon) = \frac{\int_{t_1}^{t_2} h_{unp}(t) h_{per}(t[1+\varepsilon])dt}{\sqrt{\int_{t_1}^{t_2} h^2{}_{unp}(t)dt \int_{t_1}^{t_2} h^2{}_{per}(t[1+\varepsilon])dt}} \tag{3}$$

where $\varepsilon = -\frac{\Delta V}{V}$, $t_1$ and $t_2$ are time windows, $h_{ump}$ is the unperturbed waveform, and $h_{per}$ is the perturbed waveform.

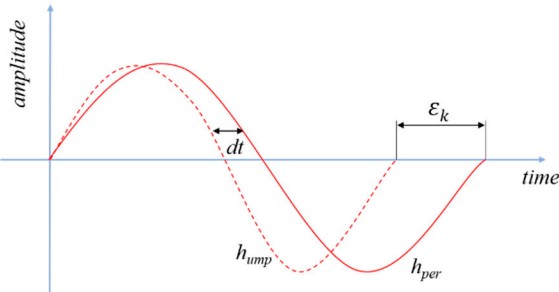

**Figure 11.** Schematic diagram of trace stretching method.

The ultrasonic wave used in the experiment is the Hanning window wave, generated by entering the following formula in the Keysight Bench Link Waveform Builder Pro software:

$$y = (\sin 2\pi ft) \times (1 - \cos(\frac{2\pi ft}{n})) \tag{4}$$

where $f$ is the frequency, $t$ is the time, and $n$ is the period.

To ensure the effective acquisition of ultrasonic waveform under different stresses during the experiments, after many tests, the sampling frequency is selected as 1 M/s, the period of the excitation wave is set to 5, and the frequency is 60 kHz. Figure 12 shows the received waveforms under different loading states.

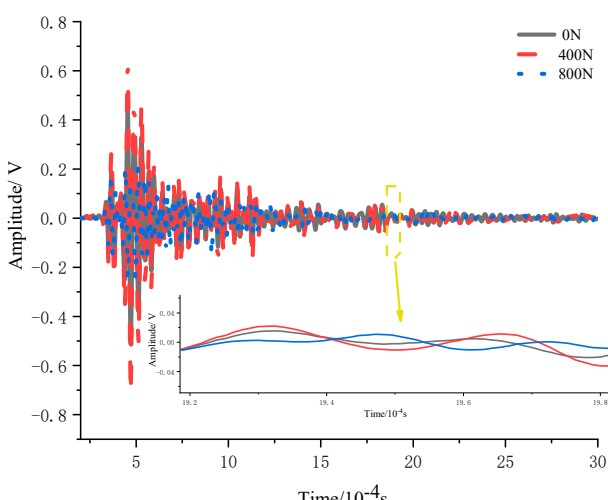

**Figure 12.** Receiving waveforms in different loading states.

### 3.2.2. Analysis of Coda Wave Velocity Variation Rate

Figure 13 shows the coda wave velocity variation rate of the LCTB with different filling times under different loading states. The figure shows that the variation rate of coda wave velocity generally tends to decrease with the increase of loading force for all three types of LCTB. From the analysis of the minimum value of the coda wave velocity variation rate of the three types of LCTB specimens, the increase in filling times makes the loading force required for the minimum wave velocity change rate smaller and the faster the coda wave velocity variation rate decreases the greater the crack extension. The internal pores of the LCTB with one filling time are relatively small compared with the other two types of LCTB. No obvious primary fractures are compacted during the loading process and new cracks are constantly sprouted, resulting in the wave velocity becoming smaller all the time. Moreover, the LCTB with two and three filling times have larger internal pores due to layered layers, obvious cracks are being compacted during the loading process, and the wave velocity increases. However, with the further increase of loading force, new cracks appear inside the LCTB and the wave velocity decreases.

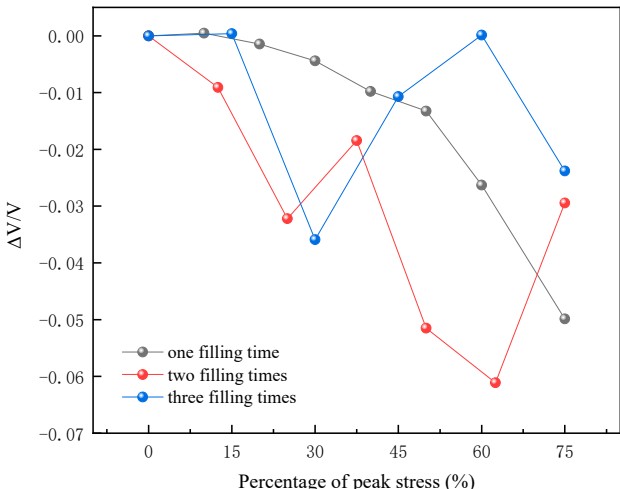

**Figure 13.** Coda wave velocity variation rate of LCTB with different filling times.

Figure 14 shows the coda wave velocity variation rate of the LCTB with different filling interval time under different loading states. It can be seen from the figure that the wave velocity variation rate of the specimen with a 0-h filling interval time decreases from 0.025 to 0. In contrast, the wave velocity variation rate of the specimen with 12-h filling interval time decreases from 0.08 to 0.04, and the wave velocity variation rate of the specimen with 24-h filling interval time decreases from 0.02 to −0.05. This phenomenon indicates that the longer the filling interval the greater the degree of a sudden decrease in the wave velocity variation rate. However, the wave velocity variation rate of all three types of LCTB can be roughly divided into three stages: up–down–up. At the beginning of the loading, the primary pore space inside the LCTB is squeezed and compacted, the pore space decreases, and the wave velocity rises. With the further increase of load, the internal sprouting of tiny cracks, the pore space increases, and the wave speed decreases.

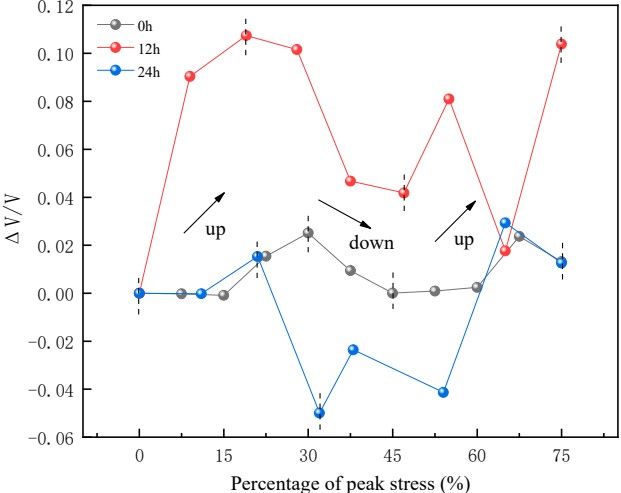

**Figure 14.** Coda wave velocity variation rate of LCTB with different filling interval time.

## 4. Conclusions

(1) With the increase in filling times, the peak stress of the LCTB decreases, the peak ringing count and cumulative ringing count decrease, and the peak energy increases. As the filling interval time increases, the stress peak of the LCTB decreases, and both the ringing count peak and energy peak decrease.

(2) With the increase of filling times, the later the minimum value of the correlation dimension *D* value of the ringing count appears, the disordered state of the LCTB occupies

the more time. The change process of the correlation dimension $D$ value of the ringing count of the LCTB with different filling times before the peak stress can be roughly divided into three stages: up–down–up.

(3) As the filling interval time increases, the minimum value of the $D$ value of the ringing count of the LCTB is smaller. The change process of the correlation dimension $D$ value of the ringing count with different filling interval time before the peak stress can be roughly divided into three stages: down–up–down.

(4) With the increase in filling times, the loading force required for the minimum value of the wave velocity variation rate of the LCTB is smaller and the faster the coda-wave velocity variation rate decreases the greater the crack expansion. The longer the filling interval time the greater the degree of the abrupt decrease in the wave velocity change rate. The change process of wave velocity variation rate of LCTB with different filling interval time can be roughly divided into three stages: up–down–up.

**Author Contributions:** C.H.: conceptualization, methodology, writing—review and editing; W.H.: supervision, conceptualization, investigation, data curation; B.L.: conceptualization, formal analysis, writing—review and editing; M.W.: conceptualization, formal analysis, writing—review and editing; S.L.: conceptualization, formal analysis, writing—review and editing; C.X.: conceptualization, formal analysis, writing—review and editing. All authors have read and agreed to the published version of the manuscript.

**Funding:** The research is funded by the National Natural Science Foundation of China (No. 51604127 and No. 51874268), China Postdoctoral Science Foundation (No. 2019M650156), Jiangxi Postdoctoral Science Foundation (No. 2018KY41), Key R&D Program of Jiangxi Province (No. 20202BBG73001) and the Science and Technology Innovation Talent Project of Ganzhou City (No. 202101094905).

**Data Availability Statement:** The data presented in this study are available on request from the corresponding author.

**Acknowledgments:** We are grateful to Jiangxi University of Science and Technology for providing us with the experimental platform and all the reviewers' specific comments and suggestions.

**Conflicts of Interest:** The authors declared that there is no conflict of interest.

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
