# Peer review of "Study on Acoustic Emission and Coda Wave Characteristics of Layered Cemented Tailings Backfill under Uniaxial Compression"

_minerals, doi:10.3390/min12070896_

Round 1
Reviewer 1 Report
1. It is suggested that add some transition sentences between the first and second sentence in Introduction.
2. Lines 49, 54, 59, etc. should be Cao et al., Cheng et al., Zhao et al. … Please correct it throughout the paper.
3. The introduction section should enrich the relationship between the manuscript and previous studies.
4. Line 88, SiO2?
5. Lines 101, 103 and 108, add space between number and unit (12 h, 24 h…).
6. Line 114, why 0.02mm/s? Please add appropriate citation.
7. Titles of right axis in Figures 4, 5, 6 and 7 are not clear.
8. In Figure 5, the legend is incorrect and it is recommended to verify and revise it.
9. For layered cemented tailings backfills with different filling interval time, why is the cement tailings ratio of the filling slurry selected inconsistently for the two stages? What is the basis for this?
10. The target stress values for each type of layered cemented tailings backfill are not directly given in the uniaxially loaded ultrasonic test program, and specific stress values are not given in the analysis of the coda wave velocity change rate under different loading states, which are suggested to be added.
11. Line 177, improve the layout of formulas.
12. Line 191, authors claim that all three types of LCTB can be divided into three stages: up-down-up. Actually, only the data of three filling times meet the trend very well.
13. Line 332, doi: 0.1016/j.conbuildmat.2021.125278.
14. Line 335, doi: 10.1016/j.ijmst.2021.01.008.
Author Response
Dear Reviewer,
Thank you for your comments.
Attached is the response.
Kind Regards

Reviewer 2 Report
The manuscript "Study on Acoustic Emission and Coda Wave Characteristics of Layered Cemented Tailings Backfill Under Uniaxial Compression" by Chongjie Huang, Wen He, Bokai Lu, Manman Wang, Shenhai Li, Changbo Xiao was submitted for review.
I read the submitted manuscript with great interest. The author turned to a very urgent problem: the creation of a backfill composite and the study of its properties, focusing on the study of acoustic emission and ultrasonic permeability of the composite during destruction.
From my point of view, the authors have done a very good study, the manuscript is replete with a large amount of demonstration material, but at the same time there are many significant and minor flaws that worsen the manuscript and reduce its quality.
From my point of view, a number of changes should be made to the manuscript that will improve its quality, enhance the ease of perception of the presented material and increase the interest of the reader.
1.) From my point of view, the title of the article is very vague, does not reflect the value and originality of this study. It follows from the title that the authors are investigating the characteristics of a certain material. It would be desirable to be more specific. The authors are doing a very good study and it would be preferable if the essence of the study would be stated in the title.
2.) From my point of view, there are very few keywords. Keywords enable the reader to quickly search for the necessary material and enable the author to popularise their research and increase interest and citations. But if this number of keywords satisfies the requirement of the journal, this comment is advisory.
3.) From my point of view, the abstract is not formed correctly. The abstract should clearly indicate the purpose of the study, its importance for society (i.e. to characterize the problem), identify the methods and materials of the study, and the conclusions should be clearly and briefly formulated. It seems that the authors have taken certain phrases from the text and thus formed the abstract.
It is inappropriate to describe the results in detail in the abstract (line 20-29), this should be placed in the section "Results and discussions". But about the methods used in the study it should be noted, for example: during the study, the authors used a multi-stage (or multi-criteria) approach based on a comparative analysis of the obtained data in the course of (specify which) observations. And the results are just as brief: a direct proportional dependence of the velocity on ... or similar. These remarks can be continued. From my point of view, the abstract must be changed.
3.1) It is desirable to avoid narrative text in the abstract.
3.2) Try to use words and phrases: an analysis has been carried out; studied; developed; proposed; established and so on. It is advisable to start sentences in the abstract with these words and phrases.
3.3) At the end of the abstract, it is necessary to indicate the final result obtained by the authors, for example: A model has been developed that allows ...; A dependence has been established which is...; A pattern has been revealed...; An efficient system (technology) has been proposed, and so on.
3.4.) Abbreviations should be avoided in the abstract. Abbreviations are allowed if it is an established international expression (for example GPS).
4.) The manuscript has a sufficient list of references, but it presents a very weak geography of citations. The list of references is intended to demonstrate the depth of the author's study of the material, the relevance and interest of their research. There are a lot of references to Chinese authors.
4.1.) The depth of study is demonstrated with the number of references - is enough;
4.2.) Relevance – with the availability of research in recent years – is enough.
4.3.) Interest – with the availability of research by scientists from different countries - is not enough.
The authors refer mainly to studies by Chinese scientists, but I would also like to point out that many scientists are working on the creation of a stowing mass on the basis of industrial waste and studying the properties of such a mass. From my point of view, the authors have unreasonably leave out the studies of Russian, Eastern European, North American scientists: Đurđevac Ignjatović, L.; Krstić, V. (Serbia); Kongar-Syuryun, Ch.B.; Ivannikov, A.; Ermolovich, Е.; Golik, V. and others (Russia); Rybak, J.; Ubysz, A; Kowalik, T and others (Poland); Bučinskas, A. (Lithuania); Kriipsalu, M. (Estonia); Grabinsky, M.; Jafari, M (Canada) and the list goes on.
In the works presented below, the authors conduct research in the field of sustainable development.
a.) Đurđevac Ignjatović, L.; Krstić, V.; Radonjanin, V.; Jovanović, V.; Malešev, M.; Ignjatović, D.; Đurđevac, V. Application of Cement Paste in Mining Works, Environmental Protection, and the Sustainable Development Goals in the Mining Industry. Sustainability 2022, 14, 7902. https://doi.org/10.3390/su14137902.
b) Khairutdinov, A.; Ubysz, A.; Adigamov, A. The concept of geotechnology with a backfill is the path of integrated development of the subsoil. IOP Conf. Ser. Earth Environ. Sci. 2021, 684, 012007. https://doi.org/10.1088/1755-1315/684/1/012007.
с) Kongar-Syuryun, Ch.; Ivannikov, A.; Khayrutdinov, A.; Tyulyaeva, Y. Geotechnology using composite materials from man-made waste is a paradigm of sustainable development. Materials Today: Proceedings 2021, 38, 2078-2082. https://doi.org/10.1016/j.matpr.2020.10.145 (In this article, the authors consider mining waste as a product and propose its use in a closed cycle of mineral extraction).
d) Kowalik, T.; Ubysz, A. Waste basalt fibers as an alternative component of fiberconcrete. Materials Today: Proceedings 2021, 38, 2055–2058. https://doi.org/10.1016/j.matpr.2020.10.140
f) Ermolovich, E.A.; Ivannikov, A.L.; Khayrutdinov, M.M.; Kongar-Syuryun, C.B.; Tyulyaeva, Y.S. Creation of a Nanomodified Backfill Based on the Waste from Enrichment of Water-Soluble Ores. Materials 2022, 15, 3689. https://doi.org/10.3390/ma15103689
g) Grabinsky, M.; Jafari, M.; Pan, A. Cemented Paste Backfill (CPB) Material Properties for Undercut Analysis. Mining 2022, 2, 103-122. https://doi.org/10.3390/mining2010007
This list can be continued. I hope the author can do it. By adding South African, Russian and Eastern European scholars to the list of reference, the authors will demonstrate an additional depth to their research, increase the percentage of international studies in the references and prove a great interest in this study. It is necessary to supplement the list of references with studies of scientists from different countries to show interest and relevance.
5.) I would recommend avoiding group references [15-23]. From my point of view, allowed up to three; more than three references are not acceptable and must be deciphered. Each paper you refer is unique and the studies you refer deserve more proper and careful review to demonstrate (and prove) its importance for the current research. It is necessary to demonstrate in detail the essence of each study and their need for your work. From my point of view, citation shows the reader to the depth of the material study and each article, to which the author refers, proves his statement. It is advisable to make references after each statement. Thus, you avoid group references and locally demonstrate to the reader the uniqueness of the link, and in general the depth of study of the material. Group references need to be separated.
6.) The authors address a very interesting topic and make a good analysis of what has been done before. But it would be desirable if the outcome of this analysis would be the authors' opinion on what is missing from the previous studies. In order to demonstrate the importance of this study, it is necessary to indicate what has not been done in previous studies or what has not been done correctly.
7.) At the end of the introduction, there is no conclusion on the analysis carried out. This conclusion allows to characterize the actual question posed, the purpose of the study and the tasks to be solved to achieve this goal. For example: Analyzing the above, it can be noted that ... is a very topical issue. Therefore, the purpose of this study is ... and to achieve this, it is necessary to solve the following tasks: 1); 2); ... Such a conclusion allows the reader to understand the vector of the study, and the authors to correctly formulate the conclusions.
8.) From my point of view, the materials are not well described. There is no information about what type of binder was used, what is the ratio between the components of the backfill: grouting fluid / binder / aggregate. Necessary to specify the brand of cement.
9.) The methods for creating backfill are not well described. From my point of view, the following should be specified:
9.1) How the materials were mixed
9.2) What instrument was used to mix; time and velocity of mixing
9.3) Due to the fact that the volumes of the backfill components are different, it is necessary to indicate how high-quality mixing and uniform distribution of the components throughout the entire volume of the composite was achieved.
10.) From my point of view, the methods of testing samples for compression are not well described. It is necessary to specify:
10.1) loading time
10.2) loading velocity
11.) From my point of view, the method of X-ray phase examination is not well described. It is necessary to specify:
11.1) The instrument on which the X-ray phase analysis was carried out
11.2) In view of the fact that industrial tailings massifs are quite large and have been forming for more than one decade, the reader needs to understand, and the author to convey to the reader, how the convergence of the X-ray phase analysis result was achieved: in what places and on what spread over the area samples were taken; these samples are averaged or several analyzes were done and then the results are averaged; if the samples are averaged, how was the mixing done and so on.
In order to better correct remarks (8)-(11) I would recommend the authors to review article (f) mentioned by me in remark (4). In this article, the authors describe quite well the methods, instruments for creating a backfill, methods for investigating the properties of the created cemented backfill.
12.) From my point of view, the conclusions are very blurred. Conclusions should be short and concise: Installed dependency...; proven possibility; and so on. What the authors describe in the conclusions is more relevant to the discussion section.
13.) It seemed to me that the authors slightly underestimated the fundamental nature of their study, due to the fact that they did not derive the dependence of "Acoustic Emission and Coda Wave" on the strength of the backfill being created. This dependence will allow to explore the fill mass in underground conditions without core sampling.
Resume: From my point of view, the authors conducted an excellent scientific study on a fairly topical issue. In general, the manuscript is of scientific and practical interest and therefore, in my opinion, can be published in the open press. The authors need to eliminate the minor comments indicated above.
Author Response
Dear reviewer,
Thanks for your comments on the paper.
The attached file is the responses to the comments.

Round 2
Reviewer 2 Report
The manuscript "Study on Acoustic Emission and Coda Wave Characteristics of Layered Cemented Tailings Backfill Under Uniaxial Compression" by Chongjie Huang, Wen He, Bokai Lu, Manman Wang, Shenhai Li, Changbo Xiao was submitted for re-review.
I am completely satisfied with the response of the authors and the changes made. From my point of view, the authors did a lot of additional work and eliminated the shortcomings that affected the quality of the manuscript.
From my point of view, the manuscript in a modified form can be published in the open press.
This manuscript is a resubmission of an earlier submission. The following is a list of the peer review reports and author responses from that submission.